# Critical Factors Affecting Outcomes of Endodontic Microsurgery: A Retrospective Japanese Study

**DOI:** 10.3390/dj12080266

**Published:** 2024-08-20

**Authors:** Masashi Yamada, Norio Kasahara, Satoru Matsunaga, Rie Fujii, Norihito Miyayoshi, Sayo Sekiya, Isabel Ding, Christopher A. McCulloch

**Affiliations:** 1Department of Endodontics, Tokyo Dental College, Kanda Misaki-cho, Chiyoda-ku, Tokyo 101-0061, Japan; myamada21@tdc.ac.jp (M.Y.); rifujii@tdc.ac.jp (R.F.); miyayoshinorihito@tdc.ac.jp (N.M.); 2Department of Histology & Developmental Biology, Tokyo Dental College, Kanda Misaki-cho, Chiyoda-ku, Tokyo 101-0061, Japan; 3Department of Anatomy, Tokyo Dental College, Kanda Misaki-cho, Chiyoda-ku, Tokyo 101-0061, Japan; 4Faculty of Dentistry, University of Toronto, 124 Edward St., Toronto, ON M5G 1G6, Canada; isabel.ding@utoronto.ca (I.D.); christopher.mcculloch@utoronto.ca (C.A.M.)

**Keywords:** endodontic microsurgery, root canal treatment, isthmus preparation, short-term outcomes

## Abstract

The critically important preoperative and intraoperative factors that affect the success of endodontic microsurgery (EMS) in Japanese patients are not defined. We conducted a retrospective study that analyzed treatment outcomes for 46 teeth in 46 Japanese patients. Treatment was provided between March 2013 and March 2015. All patients were evaluated after one year, the shortest time period over which treatment outcomes after apicoectomy could be evaluated and in which there were complete records for the recruited patient population. Healing was assessed on the basis of clinical symptoms and radiographs. With the use of a binary logistic regression model to quantify success, we estimated the effects of patient age, sex, dental arch, lesion size, lesion type, preoperative root canal treatment, the presence or absence of a post core, and the presence or absence of an isthmus on the surgically prepared dentine surface. The overall success for EMS was 93.5% after one year; failures comprised 6.5%. Successful outcomes were higher (*p* = 0.04) for maxillary teeth than for mandibular teeth. Success was higher (*p* = 0.019) for patients who received root canal instrumentation prior to EMS. Age, sex, lesion size, lesion type, the presence or absence of a post core, and the presence or absence of a root canal isthmus had no effect (*p* > 0.2) on success. We conclude that the percentage of successful outcomes after EMS treatment for Japanese patients presenting with periapical periodontitis is very high after one year and that success is influenced strongly by the dental arch and preoperative root canal instrumentation.

## 1. Introduction

Non-surgical endodontic therapy is currently the most favored treatment method for clinical management of periapical periodontitis [1]. Despite positive outcomes associated with non-surgical treatments, a certain but poorly defined proportion of interventions fail [2,3,4,5,6] for an array of reasons. These reasons include the presence of residual bacteria in difficult-to-instrument root canal systems and the presence of extra-apical infection [7]. While a surgical approach is, in almost all cases, not the first choice for initial endodontic treatment, in failed cases, surgical endodontic treatment is often employed. Notably, one of the indicated surgical approaches for the clinical management of failed cases is endodontic microsurgery (EMS) performed with a microscope. This treatment approach provides higher proportions of successful outcomes than traditional root-end surgery [8]. The improved success associated with EMS may arise from the rigorous application of defined surgical procedures, more advanced instrumentation, improved retrograde root canal cavity preparation and filling materials, as well as high-resolution identification of underlying structural problems enabled by microscopy [9,10,11].

The percentage of successful outcomes following EMS is affected by several pre-treatment factors, which include the provision of root canal treatment prior to apicoectomy [12]. While previous instrumentation has been associated with improved outcomes, the placement of a post-and-core restoration has been reported to negatively affect prognosis [13]. Currently, no definitive conclusions have been reached with respect to these factors. In addition, the failure of apicoectomy has been reported to be due to insufficient apical preparation and retro filling, or the presence of an isthmus [14]. For these reasons, if an isthmus is present on the surgically prepared surface of the root, isthmus preparation may be performed, but the effect of this preparation on post-treatment success is not defined [15]. While the therapeutic value of EMS is recognized in Japan [16,17], no studies have quantified outcomes on the relative importance of the pre-treatment factors that are described above. The null hypothesis of this study is that none of these preoperative and intraoperative factors affect the success of EMS; the alternative hypothesis is that at least one of these factors affects the success of EMS with a one-year follow-up. Our aim was to conduct a retrospective analysis to test these hypotheses.

## 2. Materials and Methods

### 2.1. Study Design

This study was conducted according to the principles of the Declaration of Helsinki (2013). Written, informed consent was obtained from all patients. The study was approved by the Tokyo Dental College Institutional Review Board (Tokyo Dental College Ethical Review No. 515). A retrospective analysis of outcomes was conducted for patients who underwent microscope-guided apicoectomy in the Department of Conservative Dentistry of the Tokyo Dental College at Chiba Hospital between March 2013 and March 2015. The observation period was for one full year, which was the shortest time period during which the treatment outcome of apicoectomy could be evaluated and which enabled the inclusion of recruited patients who completely fulfilled all inclusion criteria.

### 2.2. Sample Selection

Based on the medical and dental histories, patients who fulfilled the following inclusion criteria were selected:No medical contraindications for surgical endodontic therapy (ASA-PS class I or II).Diagnosis of periapical periodontitis by preoperative cone-beam computed tomography (CBCT).Root resection and retrograde root canal filling were conducted with the aid of a surgical microscope and with specialized instruments and treatment protocols optimized for EMS.The patient provided informed consent for the treatment.

We included cases in which EMS was performed after root canal treatment and cases in which EMS was performed without prior root canal treatment. For this latter situation, we defined cases as those in which removal of the prosthesis was difficult or in which the patient did not wish to have the prosthesis removed for esthetic reasons.

Patients that exhibited root perforation, advanced marginal periodontitis, or root fracture during preoperative examinations were excluded from the study.

### 2.3. Surgical Treatment Procedures and Follow-Up

All procedures were conducted by a single, board-certified endodontist with 15 years’ experience. The same operative procedures were used for all patients enrolled in this study. Other than the performance of the incisions, procedures were conducted with a magnified view provided by a surgical microscope (OPMI PROergo, Carl Zeiss, Gottingen, Germany). Below we describe the surgical method.

Infiltration (for maxillary teeth) or block anesthesia (for mandibular teeth) was performed with 2% xylocaine with 1:80,000 epinephrine. Cognizant of the condition of the periodontal tissues and cosmetic appearance, releasing incisions were placed through the gingival sulcus and the gingival papilla. In some instances, an Ochsenbein flap design was used. A full thickness mucoperiosteal flap was reflected; low tractional forces were applied during the surgical procedure to facilitate visual inspection of the tooth root. If needed for improving visual access to the root apex, the labial surface of the alveolar bone was removed by ostectomy. This approach enabled the exposure of the apex of the root and the lesion. A tapered fissure bur in a 45° angled-head air turbine drill was used to resect the root perpendicular to its long axis within ~3 mm of the apex. After the apical portion had been severed from the main root structure, the lesion was excised and immersed in 10% formalin solution. Hemostasis was conducted by applying pressure with a cotton swab soaked in 1:000 epinephrine. The cut surface was stained with methylene blue and examined under magnification. A 3 mm deep retrograde cavity was prepared with an ultrasound tip (KIS Ultrasonic Tip, Obtura-Spartan, Fenton, MI, USA). After enlargement, the cut surface was again stained with methylene blue and examined. Upon confirming that there were no structural defects in the root, retrograde filling was conducted by inserting MTA cement (ProRoot MTA White, Dentsply Tulsa Dental, Tulsa, OK, USA). After placement of the filling, the integrity of the MTA seal was confirmed under magnification and by obtaining radiographs of the affected site.

The flap was replaced and closed with interrupted sutures (6/0 nylon). After the application of pressure and ice packs for 10 min, hemostasis was confirmed. All patients received amoxicillin hydrate (250 mg every 8 h for 3 days, starting immediately after the procedure) and were provided with either an anti-inflammatory analgesic (Loxoprofen sodium hydrate 60 mg) or Acetaminophen (500 mg) every 6 h after surgery. The excised lesion was submitted for histopathological examination by a board-certified pathologist. The wound was disinfected with chlorhexidine. The sutures were removed on day 4. Patients who underwent surgery were contacted 12 months later and re-examined for an assessment of healing. Patients who could not be contacted and those who declined to undergo re-examination were considered as lost to follow-up. At the 12 months examination, the presence or absence of signs and symptoms was evaluated, and dental radiographs were obtained. Periapical dental radiographs were obtained in such a manner that the positioning of the radiograph holder reproduced the same position as was obtained immediately after surgery.

### 2.4. Data Collection and Outcome Assessment

The following data were obtained from the patients’ records using the parameters identified below and recorded:

The age at surgery, sex, dental arch (maxillary or mandibular), lesion size, lesion type, preoperative root canal treatment, presence or absence of a post core, and presence or absence of an isthmus on the cut surface were recorded in the database. Preoperative and postoperative signs and symptoms (pain, and pain on percussion and palpation) and X-ray findings were also recorded. The size of the lesion was measured two dimensionally by morphometry using dental radiographs. The average of the smallest and largest diameters of the lesion was recorded as the size of the lesion.

Patient assessments were conducted using postoperative dental radiographs and the criteria of healing of periapical lesions provided earlier by Rud et al. [18] and Molven et al. [19]. Based on these criteria, the status of periapical healing was classified as: complete healing, incomplete healing, uncertain healing, or unsatisfactory healing. Of these categories, complete healing and incomplete healing were assessed as “clinical success”, (Figure 1) and uncertain healing and unsatisfactory healing were considered as “clinical failure” (Figure 2). If the tooth had more than one root, the root that exhibited the least advanced healing was assessed and recorded as that which reflected the overall status of healing for that tooth. Radiographs were evaluated by three calibrated, board-certified endodontists. If their assessments were not in agreement, they discussed the case until they reached a consensus assessment. The success rate was then calculated from this assessment.

### 2.5. Statistical Analysis

A binary logistic regression model was used in which clinical healing was the dependent variable. This measure was then used to evaluate associations between independent variables and treatment outcomes (“clinical success” versus “clinical failure”). The unadjusted odds ratios were calculated to evaluate the associations between variables. The effects of different variables on healing and their associations were evaluated using odds ratios and 95% confidence intervals.

EZR ver 1.60 (Jichi Medical University Saitama Medical Center, Saitama, Japan) was used for all statistical analysis [20]. EZR is a graphical user interface (GUI) for R ver 4.20 (The R Foundation for Statistical Computing, Vienna, Austria). More precisely, EZR is the improved version of R Commander, which was designed to provide statistical functions that are frequently used in biological statistics. For parametric data, the results are reported as mean ± standard deviation.

## 3. Results

A total of 55 patients underwent EMS procedures between March 2013 and March 2015. Of these patients, nine were excluded from the study. In two cases, this was because root fracture was diagnosed intraoperatively, and the tooth was extracted; the other seven patients could not be contacted. The re-examination rate was 83.6% (46/55), and the final analysis set included 46 teeth of 46 patients. There were more than 2.5-fold more women (*n* = 34) than men (*n* = 12). The mean age of the patients was 46.2 ± 12.5 years. According to the criteria described above for periapical healing after endodontic treatment [18,19], there was complete healing in 76.1% of cases (35/46), incomplete healing in 17.4% (8/46), uncertain healing in 2.2% (1/46), and unsatisfactory healing in 4.3% (2/46). Based on our working description of treatment success described above, clinically successful outcomes were 93.5% of cases and clinical failures comprised 6.5% of cases. Of the clinically failed cases, only one case of uncertain healing exhibited no clinical symptoms. Unsatisfactory healing was found in one case that was asymptomatic, and in a single, separate case, there was pain upon palpation over the root apex.

The mean lesion diameter was 5.2 ± 3.2 mm. Histopathological analysis showed that the excised lesions were associated with different pathological diagnoses (Table 1).

We evaluated the association between healing outcome and prognostic factors (Table 2). There were more successful outcomes for teeth in the maxillary arch than for the mandibular arch (*p* = 0.041). Treatment success was higher (*p* = 0.019) when preoperative root canal treatment had been conducted than when no preoperative root canal treatment was provided. There were no significant differences (*p* > 0.2) of treatment success with respect to age, sex, lesion size on the previous radiograph, the presence or absence of a post, or the presence or absence of a root canal isthmus.

## 4. Discussion

Previous studies have indicated that the minimum time period over which treatment outcomes after apicoectomy can be reliably evaluated is one year [18,19]. In future analyses, short-term outcomes should be compared with long-term outcomes to assess if there are marked shifts in the percentage of successful outcomes. For this reason, we examined Japanese patients who had undergone EMS one year previously for treatment of periapical periodontitis. We assessed the effects of preoperative and intraoperative factors on the percentage of successful outcomes after EMS, which was 93.5%. This result is similar to that indicated in previous reports [21,22,23]. As these earlier reports described analyses conducted on non-Japanese populations, these data collectively indicate that there are no substantial racial differences in EMS outcomes, at least with respect to Japanese populations. Furthermore, of the preoperative and perioperative factors that were analyzed, only dental arch (maxillary or mandibular) and preoperative root canal instrumentation showed statistically significant differences with respect to the % of treatment success. The other factors that were analyzed (lesion size, lesion type, presence or absence of a post core, and presence or absence of an isthmus on the cut surface) did not affect treatment success.

We found no association between age or sex and the percentage of successful outcomes after EMS. In this study, the younger and older age groups were established on the basis of the average age of the patients. A previous report indicated that successful outcomes were more prevalent for younger patients in their 20s [24], and that the percentage of failed outcomes increased with age [13]. Yet, reports show higher percentages of success for older patients [25]. Notably, other studies found no association with age and successful outcomes [26,27]. With respect to the association with sex, one previous study reported that success was higher for women [28], whereas another found that successful outcomes were higher for men [13]. A study that used an insurance database to investigate treatment outcomes after apicoectomy found that the percentage of surviving teeth was higher for men than for women, and that the percentage of surviving teeth decreased continuously with the age of the patient [29]. In contrast, other studies found no significant differences with respect to age and sex [30,31,32]. Collectively, as the published literature on the effects of age and sex on treatment outcomes are not wholly consistent with their conclusions, these factors do not seem to conclusively influence the success of EMS.

In the current study, the percentage of successful outcomes was higher for the maxillary arch than the mandibular arch. This result may have been influenced by the relatively smaller sample size of teeth analyzed for the mandibular arch. Previous data indicate that the percentage of successful outcomes is higher for the maxillary arch than for the mandibular arch [13,33] and that success for the maxillary anterior teeth is reportedly higher than for posterior teeth in the maxilla. This difference may reflect the comparatively easier access and the less complex anatomical structure for anterior teeth [34].

In the current study, similar to previous analyses [35,36,37], the effect of lesion size was investigated by dividing lesions into two groups based on dental radiography: detectable lesions with a diameter < 5 mm or lesions ≥ 5 mm. We found that the mean lesion size was 5.20 mm and was not associated with any particular healing outcome one year later. Previous reports indicated the percentage of successful outcomes was positively associated with lesions of diameter < 5 mm, while lesions with a diameter ≥ 5 mm were not associated with success [38]. In contrast, longer term studies showed that lesion size did not affect the percentage of successful outcomes [18,39]. This suggestion may reflect the notion that the time required for healing depends on lesion size [21]. Conceivably, cases with healing by scarring (which we classified as “uncertain”) may exhibit periapical radiolucencies that shrink in diameter over longer time periods.

We found no effect on lesion type on the percentage of successful outcomes. Conceivably, this result may be influenced by our use of high magnification imaging during lesion removal. This finding is in contrast to previous studies in which this part of the procedure was conducted under no or low magnification. Previous data showed that radicular granulomas (73%) were more common than radicular cysts (15%) [40], which is similar to the current study in which radicular granulomas (65%) were more common than radicular cysts (20%). In contrast, the opposite result was reported in another study in which chronic, non-specific inflammatory lesions and radicular cysts (47.3%) were more common than radicular granuloma (44.0%) [41]. Furthermore, in this same study, treatment success for the apicoectomy of teeth with cysts was higher than teeth with granulomas [41]. Conceivably, this outcome may reflect the notion that cysts, which contain epithelial cell walls, are relatively easier to remove than granulomas [42].

Preoperative root canal treatment can affect healing outcomes [7]. Similarly, we found that the percentage of successful outcomes was higher when root canal treatment was performed before EMS. This result may reflect the reduced numbers of bacteria in the root canal system prior to the initiation of EMS. In addition, preoperative treatment of the root canal aids can aid clinical diagnosis. This approach thereby enables the exclusion of teeth with root fracture or perforation, which facilitates positive treatment outcomes.

For many of the cases examined here, we found that EMS was indicated in cases exhibiting very little remaining tooth substance and in which post–core removal was challenging. For these cases, there was no significant difference in the short-term percentage of successful outcomes, a result that is consistent with other studies [8]. In this context, when treatment was repeated for teeth requiring post–core removal, vertical root fracture occurred more frequently than when EMS was performed [43]. This result could arise because post–core removal reduced the amount of dentin in the root, further reducing resistance to fracture. Conversely, when EMS was performed on teeth with ideally restored post–cores, the long-term survival rate was higher [44]. Accordingly, while not yet resolved, preoperative root canal treatment apparently confers higher risk on treatment outcomes, and surgical treatment should be the first choice for teeth for which the tooth to be treated has an existing post–core.

The amount of dentin at the apex may affect treatment outcomes. Insufficient isthmus instrumentation may be one reason for the failure of apicoectomy [14]. If an isthmus is observed, certain authors encourage prophylactic enlargement and filling [14,45]. However, one study reported that the four-year survival rate of teeth in which no isthmus was present and in which isthmus preparation was not conducted was 87.4%. This survival rate is higher than the 61.5% reported when an isthmus was present and isthmus preparation was conducted [15]. Furthermore, eight of the nine cases of failure in this study were associated with vertical root fracture after teeth had undergone isthmus preparation. These data indicate that isthmus preparation may reduce the amount of dentin remaining postoperatively and reduce resistance to fracture. Notably, a micro-CT morphological study of the maxillary first molars of Japanese patients found that, even when no isthmus was present (or only when an incomplete isthmus was observed at a position 3 mm from the mesiobuccal root canal apex), morphological structures associated with root canal treatment, including isthmuses, were located more coronally [46]. For this reason, in the current study, if an isthmus was observed during apicoectomy (*n* = 26/46), whatever its shape, prophylactic isthmus preparation was conducted. This treatment was performed with a 0.5 mm diameter ultrasound tip, while being careful to remove the minimum amount of dentin. The percentage of successful outcomes measured at one year was not affected by isthmus preparation, even though the root is weakened after isthmus preparation and the possibility of root fracture may be increased [39].

EMS reportedly has a higher short-term success rate than non-surgical endodontic therapy [47], which is consistent with the data reported here. However, similar to traditional root end surgery, the long-term success rate of EMS is lower (albeit only slightly) than other short-term estimates of treatment success [26,48]. This finding is likely related to leakage of the retrograde root canal filling over time [49]. Notably, biological and treatment factors that do not manifest during the short-term can still affect the long-term prognosis. Therefore, additional, longer-term follow-up is needed.

## 5. Conclusions

The percentage of short-term successful outcomes after EMS in Japanese patients was very high (93.5%). We conclude that the major factors that affected treatment success were preoperative root canal treatment and arch type. In contrast, sex, age, lesion size, lesion type, the presence or absence of a post core, and the presence or absence of a root canal isthmus exerted no significant effect on short-term success.

## Figures and Tables

**Figure 1 dentistry-12-00266-f001:**
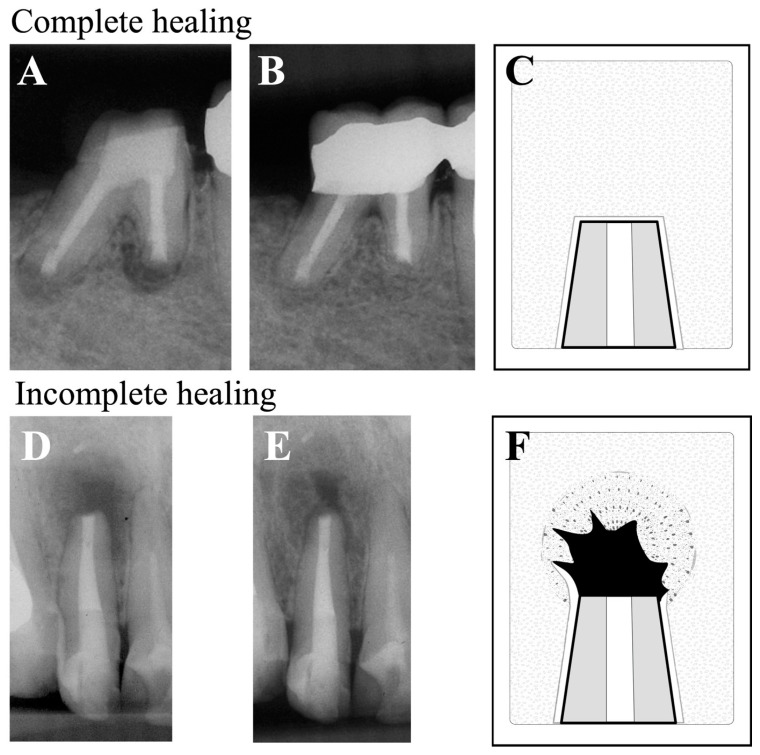
Representative radiographs as examples of clinical success. Complete healing (**Top Row**): an example of a mandibular first molar that was radiographically assessed as ‘complete healing’ 1 year after EMS. (**A**) Immediate postoperative, (**B**) 1-year follow-up, and (**C**) diagram of apical healing used for classification developed by Rud and Molven [18,19]. Incomplete healing (**Bottom Row**): a maxillary lateral incisor was radiographically assessed as an example of ‘incomplete healing’ 1 year after EMS. (**D**) Immediate postoperative, (**E**) 1-year follow-up, and (**F**) diagram of apical healing used for classification, as described [18,19].

**Figure 2 dentistry-12-00266-f002:**
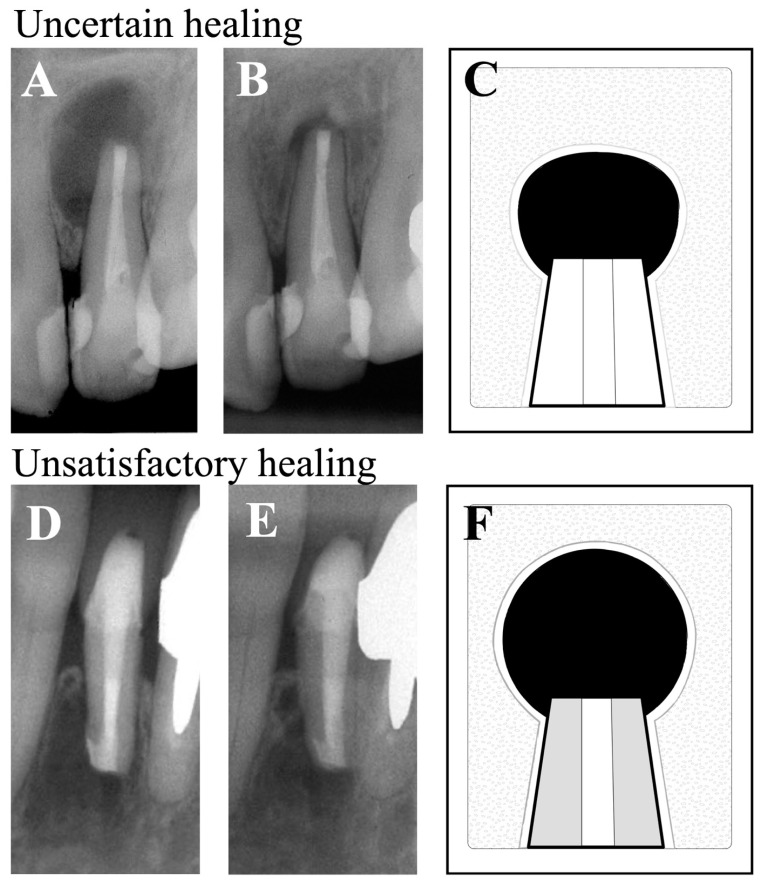
Representative radiographs as examples of unsatisfactory healing. Uncertain healing (**Top Row**): a maxillary central incisor was radiographically assessed as an example of ‘uncertain healing’ 1 year after EMS. (**A**) Immediate postoperative, (**B**) 1-year follow-up, and (**C**) diagram of apical healing used for classification, as described [18,19]. Unsatisfactory healing (**Bottom Row**): a mandibular lateral incisor was radiographically assessed as ‘unsatisfactory healing’ 1 year after EMS. (**D**) Immediate postoperative, (**E**) 1-year follow-up, and (**F**) diagram of apical healing used for classification, as described [18,19].

**Table 1 dentistry-12-00266-t001:** Types of lesions removed during EMS.

Pathological Diagnosis	*n* (%)
Radicular cyst	26 (65)
Radicular granuloma	8 (20)
Inflammatory granulation tissue	6 (7.5)
Not collected	6 (7.5)

**Table 2 dentistry-12-00266-t002:** Distribution of cases by factor and bivariate analysis.

Variable/Factor (*n*)		Clinical Success (%)	Clinical Failure (%)	OR (95%CI)	*p* Value
Sex (46)	Female	31 (67.4)	3 (6.5)	0	0.557
	Male	12 (26.1)	0 (0)	(0–7.012)	
Age, y (46)	<45	29 (63.1)	2 (4.3)	1.035	1
	≥45	14 (30.4)	1 (2.2)	(0.02–21.50)	
Arch (46)	Maxillary	39 (84.8)	1 (2.2)	17.08	0.041 *
	Mandibular	4 (8.7)	2 (4.3)	(0.75–1165.82)	
Lesion size, mm (46)	<5	17 (37)	0 (0)	Inf	0.524
	≥5	27 (58.7)	2 (4.3)	(0.11–Inf)	
Lesion type (34)	Granuloma	24 (70.6)	2 (5.9)	3.27	0.432
	Cyst	7 (20.6)	1 (2.9)	(0.04–279.35)	
Pre-RCT (46)	Yes	33 (71.8)	0 (0)	Inf	0.019 *
	No	10 (21.7)	3 (6.5)	(1.14–Inf)	
Post core (46)	Yes	17 (37)	2 (4.3)	0.561	0.561
	No	26 (56.5)	1 (2.2)	(0.01–6.91)	
Isthmus (46)	Yes	18 (39.1)	2 (4.3)	0.368	0.572
	No	25 (54.4)	1 (2.2)	(0.01–7.58)	

* Indicates significance (*p* < 0.05); CI, 95% confidence interval.

## Data Availability

The data presented in this publication are available on request from the corresponding author. The data are not publicly available due to privacy restrictions.

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
