# Peer review of "Critical Factors Affecting Outcomes of Endodontic Microsurgery: A Retrospective Japanese Study"

_dentistry, 2024, doi:10.3390/dj12080266_

Round 1

Reviewer 1 Report (Previous Reviewer 1)

Comments and Suggestions for Authors

Revisions made by the authors are fine.

Author Response

For research article

Response to Reviewer 1 Comments

1. Summary

2. Questions for General Evaluation

Reviewer’s Evaluation

Response and Revisions

Does the introduction provide sufficient background and include all relevant references?

Yes

Are all the cited references relevant to the research?

Yes

Is the research design appropriate?

Yes

Are the methods adequately described?

Yes

Are the results clearly presented?

Yes

Are the conclusions supported by the results?

Yes

3. Point-by-point response to Comments and Suggestions for Authors

4. Response to Comments on the Quality of English Language

5. Additional clarifications

Reviewer 2 Report (Previous Reviewer 2)

Comments and Suggestions for Authors

Dear authors,

thank you very much for this paper focusing on a relevant point and discussion in endodontics. Compared to the first submission the paper is improved. However, major points remain unclear.

Abstract: All patients were treated between 2013 and 2015: Why did you publish only the one-year results in 2024. A prolonged observation time seems to be possible. Please explain this point. 

Introduction: A surgical approach is in almost all cases not the choice for an initial endodontic treatment. Please add and clarify.

The impact of apical preparation and filling is one of the most important factors related to apical surgery in endodontics. Please address this missing point in the introduction section.

Please give a clear aim and hypothesis of your work. This is still missing

Material and Methods: Please provide more information about the selection of the patients for your study. Why did you present only the 12 months follow-up? Any reason? Please give more information. This is still missing. Please add.

Regarding the follow-up: A radiograph was taken. Please give more information about this point. At the beginning also CBT´s were performed. Did you use additional radiographs? Did you use any device to get comparable x-rays? Please explain. This point is still unclear.

The dimension of apical lesions is mentioned in the results section. I think CBTs were used prior the surgery procedure. Please clarify the measurements of the lesions and include this point in the M&M section. Did you use the CBTs or 2-dimensional standard radiographs? Please add and explain. This might help to understand the dimensional aspects of your work. this point is still missing.

Discussion: Please explain the fact that 2024 the 12 months results from a study performed in 2013-2015 are presented. This remains still unclear. Just the explanation that a minimum time period of one year is necessary might be correct, but your data allow longer observation times. Please clarify.

Comments on the Quality of English Language

Moderate editing is necessary.

Author Response

For research article

Response to Reviewer 2 Comments

1. Summary

2. Questions for General Evaluation

Reviewer’s Evaluation

Response and Revisions

Does the introduction provide sufficient background and include all relevant references?

Must be improved

We have described the modifications in Comment 1.

.

Are all the cited references relevant to the research?

Must be improved

We have described the modifications in Comment 2.

Is the research design appropriate?

Can be improved

We have described the modifications Comments 3.

Are the methods adequately described?

Can be improved

We have described the corrections in Comment 3.

Are the results clearly presented?

Can be improved

We have described the corrections in comment 4.

Are the conclusions supported by the results?

Can be improved

3. Point-by-point response to Comments and Suggestions for Authors

Comment 1:

・Abstract: All patients were treated between 2013 and 2015: Why did you publish only the one-year results in 2024. A prolonged observation time seems to be possible. Please explain this point. 

Response 1:

・This approach was followed as the study was conducted over a one-year period, which is the shortest period over which treatment results can be evaluated and in which full data sets consistent with the inclusion criteria could be obtained. This above information has been added to the text (lline 18-20.

“All patients were evaluated after one year, the shortest time period over which treatment outcomes after apicoectomy could be evaluated and in which there were complete records for the recruited patient population.”

Comment 2: Introduction:

a. Introduction: A surgical approach is in almost all cases not the choice for an initial endodontic treatment. Please add and clarify.

b. The impact of apical preparation and filling is one of the most important factors related to apical surgery in endodontics. Please address this missing point in the introduction section.

c. Please give a clear aim and hypothesis of your work. This is still missing.

Response 2:

a. We added in the text in the Introduction section that surgical endodontic treatment is not the first option to be considered at the start of treatment. The revised text that considers this issue has been added. (41-42)

- “While a surgical approach is in almost all cases, not the first choice for initial endodontic treatment, in failed cases, surgical endodontic treatment is often employed.”

b. As the reviewer indicated, we have indicated in the text that the success of surgical endodontic treatment is influenced by a number of factors including the nature of the apical preparation and filling. The revised text that considers this issue has been added. (55-56)

The following statement has been added to the revised text:

- “failure of apicoectomy has been reported to be due to insufficient apical preparation and retro filling, or the presence of an isthmus[14]. For these reasons,…….”

c. The following sentence was added to clarify the purpose of the study.

The revised text that considers this issue has been added. (62-63), The following revised text has been added:  “ In particular, we examined how the conduct of preoperative root canal treatment affects treatment outcomes. We hypothesized that EMS improves success of endodontic re-treatment.”

Comment 3: Material and Methods:

a. Please provide more information about the selection of the patients for your study. Why did you present only the 12 months follow-up? Any reason? Please give more information. This is still missing. Please add.

b. Regarding the follow-up: A radiograph was taken. Please give more information about this point. At the beginning also CBT´s were performed. Did you use additional radiographs? Did you use any device to get comparable x-rays? Please explain. This point is still unclear.

c. The dimension of apical lesions is mentioned in the results section. I think CBTs were used prior the surgery procedure. Please clarify the measurements of the lesions and include this point in the M&M section. Did you use the CBTs or 2-dimensional standard radiographs? Please add and explain. This might help to understand the dimensional aspects of your work. this point is still missing.

Response 3:

a. Detailed information on the selection of patients and inclusion criteria is provided in the revised text (lines 79-93). The reason for the restriction of the study period to one year is that this was the shortest time-period over which treatment results could be accurately evaluated in patients with complete data sets and which provided a sufficiently long period to ensure that the outcomes were clinically meaningful. The revised text that considers this issue has been added. (72-75)

b. No additional CBCTs were taken. Therefore, we modified by text by adding that periapical dental radiographs were taken and that the consistency of the positioning was improved with the use of an indicator. The revised text that considers this issue has been added. (130-132)

- “Periapical dental radiographs were obtained in such a manner that the positioning of the radiograph holder reproduced the same position as was obtained immediately after surgery.”.

c. In this study, a two-dimensional evaluation was performed using dental radiographic images. We have also added a description of this method. The revised text that considers this issue has been added. (140-141)

- The size of the lesion was measured two dimensionally by morphometry using dental radiographs. The average of the smallest and largest diameters of the lesion was recorded as the size of the lesion.

Comment 4: Discussion:

Please explain the fact that 2024 the 12 months results from a study performed in 2013-2015 are presented. This remains still unclear. Just the explanation that a minimum time period of one year is necessary might be correct, but your data allow longer observation times. Please clarify.

Response 4: Discussion:
The restriction of the time-period to one year is because this is the shortest period over which treatment results can be evaluated with certainty and in which all inclusion criteria could be fulfilled for all participating patients. Notably, we are planning to investigate longer-term success rates and compare these data with the current results. Accordingly, we have added a note to the text to indicate this point.

The revised text that considers this issue has been added. (220-222)

- In future analyses, short-term outcomes should be compared with longer-term outcomes to assess whether there are marked shifts in the percentage of successful outcomes.

4. Response to Comments on the Quality of English Language

Point 1: Moderate editing is necessary.

Response 1: We have carefully reviewed the entire manuscript and revised these for any grammatical or syntax errors, which are noted in the highlighted text.

5. Additional clarifications

Reviewer 3 Report (Previous Reviewer 3)

Comments and Suggestions for Authors

Thank you for the opportunity to re-read the article. As I had previously suggested, the article has the correct structure and is clear methodologically. The results are presented, and the discussion seems complete to me. The conclusion is short and straightforward. 

However, I recommend that Table 1 be formatted in the same way (size and font) as the rest of the work.

Author Response

For research article

Response to Reviewer 3 Comments

1. Summary

2. Questions for General Evaluation

Reviewer’s Evaluation

Response and Revisions

Does the introduction provide sufficient background and include all relevant references?

Yes

Are all the cited references relevant to the research?

Yes

Is the research design appropriate?

Yes

Are the methods adequately described?

Yes

Are the results clearly presented?

Yes

Are the conclusions supported by the results?

Yes

3. Point-by-point response to Comments and Suggestions for Authors

Comments 1:

・The reviewer suggested that: “I recommend that Table 1 be formatted in the same way (size and font) as the rest of the work.

Response 1:

・Cognizant of these suggestions, We have formatted Table 1 using the same size and type of font as the rest of the manuscript.

4. Response to Comments on the Quality of English Language

5. Additional clarifications

Reviewer 4 Report (Previous Reviewer 4)

Comments and Suggestions for Authors

Congratulations to the authors who achieved a great clinical scientific article in the endodontic field. The article was well-edited, with no obvious grammar errors. The only problem I was concerned about was the sample size was not big, with less scientific evidence. Considering the methods of collecting samples ( via microsurgery), it can be acceptable due to the difficulties of sample collection. The statistical method used in this research fits the design of experiments. Only the strength and significance showed less evidence if the sample size was bigger. 

The plagiarism checked with software showed less than 7%. The English editing is acceptable in this version, so this paper can be considered to be published in this version. Congratulations to the authors.

Author Response

For research article

Response to Reviewer 4 Comments

1. Summary

2. Questions for General Evaluation

Reviewer’s Evaluation

Response and Revisions

Does the introduction provide sufficient background and include all relevant references?

Yes

Are all the cited references relevant to the research?

Can be improved

We checked the accuracy of the references with a reference manager.

Is the research design appropriate?

Yes

Are the methods adequately described?

Yes

Are the results clearly presented?

Yes

Are the conclusions supported by the results?

Yes

3. Point-by-point response to Comments and Suggestions for Authors

Comments 1:

・The reviewer said that: “The only problem I was concerned about was the sample size was not big, with less scientific evidence. Considering the methods of collecting samples ( via microsurgery), it can be acceptable due to the difficulties of sample collection.

The statistical method used in this research fits the design of experiments. Only the strength and significance showed less evidence if the sample size was bigger.

The plagiarism checked with software showed less than 7%. The English editing is acceptable in this version, so this paper can be considered to be published in this version.

Response 1:

・We appreciate the reviewer's comments.

4. Response to Comments on the Quality of English Language

5. Additional clarifications

Round 2

Reviewer 2 Report (Previous Reviewer 2)

Comments and Suggestions for Authors

Dear authors,

thank you very much for improving this paper focusing on a relevant point and discussion in endodontics. Compared to the previous submission the paper is improved. 

Abstract: Has been adjusted. Thank you. 

Introduction: Thank you for improving the introduction. Information about the impact of the used filling material for root end fillings could be added. 

A clear aim, hypothesis and null-hypothesis are still missing. Please add.

Material and Methods: Thank you for improving the paper.

Discussion:  Thank you for improving the paper.

Conclusion: Please clarify the conclusion: The major fact.... remains unclear.

Comments on the Quality of English Language

Minor editing is needed.

Author Response

For research article

Response to Reviewer 2 Comments

1. Summary

2. Questions for General Evaluation

Reviewer’s Evaluation

Response and Revisions

Does the introduction provide sufficient background and include all relevant references?

Can be improved

We have described the modifications in Comment 1.

.

Is the research design appropriate?

Can be improved

We have described the modifications Comments 1.

Are the methods adequately described?

Yes

Are the results clearly presented?

Yes

Are the conclusions supported by the results?

Can be improved

We have described the corrections in Comment 2.

3. Point-by-point response to Comments and Suggestions for Authors

Comment 1: Aim・A clear aim, hypothesis and null-hypothesis are still missing. Please add.

Response 1:

・We have now provided a null and alternative hypotheses and an aim. This information has been added to the red-highlighted text. (line 61-64). “The null hypothesis of this study is that none of these preoperative and intraoperative factors affect the success of EMS; the alternative hypothesis is that at least one of these factors affect the success of EMS with a one-year follow-up. Our aim was to conduct a retrospective analysis to test these hypotheses.”

Comment 2: Conclusion ・Please clarify the conclusion: The major fact.... remains unclear.

Response 2: We have now stated our conclusions, which are in accordance with our aim. This information has been added to the red highlighted text (lines 325-328). “We conclude that the major factors that affected treatment success were preoperative root canal treatment and arch type. In contrast, sex, age, lesion size, lesion type, the presence or absence of a post core, and the presence or absence of a root canal isthmus, exerted no significant effect on short-term success.”

4. Response to Comments on the Quality of English Language

Point 1: Moderate editing is necessary.

Response 1: We have carefully reviewed the entire manuscript for any grammatical or syntax errors, and the corrections of these errors are noted in the highlighted text.

5. Additional clarifications

This manuscript is a resubmission of an earlier submission. The following is a list of the peer review reports and author responses from that submission.

Round 1

Reviewer 1 Report

Comments and Suggestions for Authors

This retrospective study presents important data for judging the outcomes of endodontic microsurgery.

However, there are some major and minor issues that I would like to address:

Major issues:

- In the methods, the authors state: "At the 12 months examination, the presence or absence of signs and symptoms was evaluated, and dental radiographs were obtained." However, the authors based their judgement whether a treatment was successful  entirely on radiographic data. In my opinion, the clinical data (pain to percussion, palpation, etc.) is equally (or more?) important for evaulating endodntic success. Therefore, I encourage the authors to add clinical data prior to EMS and post treatment. Since this study is based on clinical reports, these data should be available and easy to include into the paper.

-  I am wondering why the authors only showed the 12 months follow up data. Since the patients included in this study were treated approx. 10 years ago, I would expect at least some data after 3-5 years, which would increase the scientific relevance of this study. After all, 12 months is not much when speaking of clinical success in endodontics and the good outcomes after this short period of time are also not very surprising.

Minor issues:

- the authors stratified the variable "age" into two groups. Why was 45 years chosen as threshold age? Please explain in the discussion.

- l. 207-211. Please elaborate which age was used for defining "older" and "younger" patients in the studies cited. This is important for comparing other studies too the present one.

- l.220-227. Please do not draw any conclusions about success rates for maxillary vs. mandibular teeth from your data, since the sample size is too different in both groups. 6 teeth in total are not enough for judging the success rate of mandibular teeth.

Comments on the Quality of English Language

Minor editing required (typos, reptitions)

Reviewer 2 Report

Comments and Suggestions for Authors

Dear authors,

thank you very much for this paper focusing on a relevant point and discussion in endodontics.

Abstract: All patients were treated between 2013 and 2015: Why did you publish only the one-year results in 2024. A prolonged observation time seems to be possible. Please think about this point. The clinical recommended procedure is always a root canal therapy prior to surgical procedures. Why did you include patients without any RCT? Please explain.

Introduction: I think in your paper and the first section of the introduction you are talking about endodontic retreatment cases. Please include the fact that the paper is dealing with retreatment cases in the introduction section. Please clarify. A surgical approach is in almost all cases not the choice for an initial endodontic treatment.

The impact of apical preparation and filling is one of the most important factors related to apical surgery in endodontics. Please address this point also in the introduction section.

Please give a clear aim and hypothesis of your work.

Material and Methods: Please provide more information about the selection of the patients for your study. Why did you present only the 12 months follow-up? Any reason? Please give more information.

Regarding the follow-up: A radiograph was taken. Please give more information about this point. At the beginning also CBT´s were performed. Did you use additional radiographs? Please clarify the text.

Results: Please explain the results from the table in more detail. The indication for EMS should also be mentioned.

The dimension of apical lesions is mentioned in the results section. I think CBTs were used prior the surgery procedure. Please clarify the measurements of the lesions and include this point in the M&M section. Did you use the CBTs or 2-dimensional standard radiographs? Please add and explain. This might help to understand the dimensional aspects of your work.

Discussion: Please explain the fact that 2024 the 12 months results from a study performed in 2013-2015 are presented. This remains unclear. A longer observation time might have been possible. Please clarify.

Comments on the Quality of English Language

English is almost fine. Minor editing will be necessary.

Reviewer 3 Report

Comments and Suggestions for Authors

I am grateful for the opportunity to read the article "Critical factors affecting outcomes of endodontic microsurgery: A retrospective Japanese study," submitted to Dentistry Journal MDPI. 

The article is well structured and well presented. 

The methodology is clear and complete, the results are well-explored, and the discussion is concrete. 

My only recommendation is that the symbol % should be written as a percentage for formality in the conclusion. 

The limitations are addressed throughout the text; however, there could be a sentence that summarizes them in the discussion.

Best Regards

Reviewer 4 Report

Comments and Suggestions for Authors

There are several grammar and spelling errors in this article, which make it less valuable for publication. I personally suggest the authors consider editing proof or rechecking the editing errors in this article. The content is good. After correcting the editing errors and re-submitting the article, the article may have value for peer review. 

Comments on the Quality of English Language

The English writing needs major revise. Editing errors should be correct. I suggest the authors may check the English editing or sent for English editing proof to improve the quality of writing.